# Highly Branched Neo-Fructans (Agavins) Attenuate Metabolic Endotoxemia and Low-Grade Inflammation in Association with Gut Microbiota Modulation on High-Fat Diet-Fed Mice

**DOI:** 10.3390/foods9121792

**Published:** 2020-12-03

**Authors:** Alicia Huazano-García, María Blanca Silva-Adame, Juan Vázquez-Martínez, Argel Gastelum-Arellanez, Lino Sánchez-Segura, Mercedes G. López

**Affiliations:** 1Departamento de Biotecnología y Bioquímica, Centro de Investigación y de Estudios Avanzados del IPN, Unidad Irapuato, Guanajuato 36821, Mexico; allizhia@gmail.com (A.H.-G.); blanca.silva@cinvestav.mx (M.B.S.-A.); jvazmar87@hotmail.com (J.V.-M.); 2Cátedra CONACYT, Centro de Innovación Aplicada en Tecnologías Competitivas A.C. (CIATEC AC), León 37545, Guanajuato, Mexico; a.gastelum@conacyt.mx; 3Departamento de Ingeniería Genética, Centro de Investigación y de Estudios Avanzados del IPN, Unidad Irapuato, Guanajuato 36821, Mexico; lino.sanchez@cinvestav.mx

**Keywords:** agavins, branched neo-fructans, prebiotics, obesity, endotoxemia, microbiota, metabolites

## Abstract

Highly branched neo-fructans (agavins) are natural prebiotics found in Agave plants, with a large capacity to mitigate the development of obesity and metabolic syndrome. Here, we investigated the impact of agavins intake on gut microbiota modulation and their metabolites as well as their effect on metabolic endotoxemia and low-grade inflammation in mice fed high-fat diet. Mice were fed with a standard diet (ST) and high-fat diet (HF) alone or plus an agavins supplement (HF+A) for ten weeks. Gut microbiota composition, fecal metabolite profiles, lipopolysaccharides (LPS), pro-inflammatory cytokines, and systemic effects were analyzed. Agavins intake induced substantial changes in gut microbiota composition, enriching *Bacteroides*, *Parabacteroides*, *Prevotella*, *Allobaculum*, and *Akkermansia* genus (LDA > 3.0). l-leucine, l-valine, uracil, thymine, and some fatty acids were identified as possible biomarkers for this prebiotic supplement. As novel findings, agavins supplementation significantly decreased LPS and pro-inflammatory (IL-1α, IL-1β, and TNF-α; *p* < 0.05) cytokines levels in portal vein. In addition, lipid droplets content in the liver and adipocytes size also decreased with agavins consumption. In conclusion, agavins supplementation mitigate metabolic endotoxemia and low-grade inflammation in association with gut microbiota regulation and their metabolic products, thus inducing beneficial responses on metabolic disorders in high-fat diet-fed mice.

## 1. Introduction

The incidence of obesity worldwide has increased drastically in the last decade reaching pandemic proportions and significantly contributing to reducing the life quality of people. Environmental factors, such as high-fat diet consumption and a sedentary lifestyle are the main causes of this dramatic obesity increment [1]. However, the main problem of obesity is that this pathology is always associated with an array of additional health problems, including increased risk of fatty liver disease, insulin resistance, type 2 diabetes, atherosclerosis, degenerative disorder including dementia, airway disease, and some cancers [2]. Even presently, obese people have been among the most severely affected by the emerging chronic communicable diseases as SARS-CoV-2.

Obesity is characterized by a low-grade inflammation whose molecular origin remains unknown [3]. Nevertheless, in several mice studies it has been reported that high-fat diet consumption end up in gut dysbiosis (alterations in gut microbiota composition), commonly defined by an increase of Firmicutes and decrease of Bacteroidetes [4]; this gut dysbiosis leads to lipopolysaccharide (LPS, important constituent of the outer membrane of Gram-negative bacteria) release to the bloodstream, due to increase intestinal permeability, this process has been referred to as metabolic endotoxemia and related to the development of low-grade inflammation [5,6].

On the other hand, LPS themselves lead to the synthesis of pro-inflammatory cytokines such as tumor necrosis factor (TNF)-α, interleukin (IL)-1, and (IL)-6 by activation of the toll-like receptor 4 (TLR4) [7,8], which is a key regulator of the innate immune response to bacteria and has been proposed to act as a connection between inflammation and metabolism [9]. In addition, the increment of LPS and pro-inflammatory cytokines in peripheral blood could have profound effects on liver and adipose tissue [6,10]; promoting the development of non-alcoholic fatty liver (NAFL) and an augment of adipocytes size [11,12,13], which also plays crucial roles in the development of metabolic syndrome via multiple mechanism including impaired insulin signaling and low-grade inflammation [2].

In the context of low-grade inflammatory state, prebiotics such as inulins (linear fructans) reduce metabolic endotoxemia through reduction of intestinal permeability, derived from gut microbiota modulation and the enrichment of probiotic bacteria such as *Bifidobacterium* spp. and *Akkermansia muciniphila* [10,14]. On the other hand, recently, increasing interest in highly branched prebiotics has arisen from recent findings including the modulation of gut microbiota composition driven by agavins intake, particularly decreasing the Firmicutes/Bacteroidetes ratio and promoting the increased abundance of the Enterobacteriaceae family, with positive effects on body weight loss in overweight mice [15]. In addition, agavins intake also mitigate the metabolic syndrome, through the generation of metabolic products and satiety hormones such glucagon-like peptide-1 (GLP-1) and leptin [16,17].

Agavins possess a complex structures and very unique features due to the presence of an internal glucose unit as well as fructose molecules linked by common β(2-1) like inulins plus β(2-6) linkages [18]; therefore, they cannot be degraded by endogenous gastrointestinal enzymes during their passage through the stomach and small intestine; thus, they reach the colon structurally unchanged, and they are fermented by gut microbiota present in the colon [15]. The fermentation of complex fructans, such as agavins, involves the collaboration of a highly diverse selection of gut microbes (including probiotic bacteria), which produce a myriad of different metabolites that are suggested as key links in the communication between bacterial communities of the gut and the host [19]. In this sense, fecal metabolites analysis can provide a non-invasive manner to study the outcome of the host–gut microbiota interaction and open an important option to propose new biomarkers (different to SCFA) with new therapeutic approaches, through selective alterations of microbial production molecules to promote host health and prevent diseases [20].

Moreover, substantial growing evidence has demonstrated agavins’ efficacy on the reduction of hepatic steatosis [21,22]. However, until now, there are no studies showing an integrative way by which agavins impact gut microbiota modulation and their metabolic products, nor the effect that they have on metabolic endotoxemia and low-grade inflammation—both of which play a key role on the development of the metabolic syndrome in obese individuals [2,3]. Therefore, in the present study, we investigated the agavins’ capacity to reduce metabolic endotoxemia as well as low-grade inflammation and their impact on the metabolic syndrome in obese mice. Our hypothesis was that agavins supplementation, through the gut microbiota modulation and their metabolites, might lead to the reduction of circulating LPS, mitigating liver and adipose tissue damage with beneficial consequences on metabolic disorders of obese mice. The identification/characterization of metabolites generated from the microbial fermentation of agavins opens opportunities to transform this information into practical solutions to decrease obesity.

## 2. Materials and Methods

### 2.1. Animals and Diets

Twenty-one male C57BL/6 mice (8 weeks old mice at the beginning of the experiment were obtained from the Universidad Autonoma Metropolitana, Mexico City, Mexico) and individually hosed in a separate cage, but in the same room, with temperature and humidity controlled and 12 h light/12 h dark cycle (07.00 a.m.–19.00 p.m. light period). All animals were adapted for one week upon arrival before subsequent experiments. Mouse were randomized divided to one of the three experimental diets (*n* = 7 per group): a control group (ST) was fed with a standard diet (5053 Lab Diet, St. Louis, MO, USA); and two high fat groups were fed with a high-fat diet (58Y1 Test Diet, Richmond, IN, USA) alone (HF) or with an agavins supplement in their drinking water (HF+A). The standard diet (5053 Lab Diet) contained 62.4% calories from carbohydrates (28.6% starch, 3.24% sucrose, 1.34% lactose, 0.24% fructose, and 0.19% glucose); 24.5% from proteins and 13.1% from fat; whereas the high-fat diet (58Y1 Test Diet) had 20.3% calories from carbohydrates (16.15% maltodextrin, 8.85% sucrose, and 6.46% powdered cellulose), 18.1% from proteins, and 61.6% from fat (31.7% lard and 3.2% soybean oil).

Agavins from *Agave tequilana* (BIOAGAVE^TM^ powder; code: 11200001) were obtained from Ingredion Mexico. Each 100 g of this product contains 91.6 g of agavins (soluble fiber) and 2.8 g of sugars as well as 5.6% of moisture, according to information provided by the supplier. We analyzed BIOAGAVE^TM^ powder by high-performance anion-exchange chromatography with pulsed amperometric detection (HPAEC-PAD; Dionex ICS-3000, Sunnyvale, CA, USA) obtaining a characteristic chromatographic profile for branched fructans (Appendix A). Agavins were added in bottled water at a concentration of 0.38 g/mouse/day. The agavins were freshly prepared daily by dissolving 8.141 g of BIOAGAVE^TM^ powder in 105 mL of water. Clean drinking vessels were then filled with an equal volume of the solution and gave to the seven cages of HF + A group. Water intake was monitored daily to ensure all animals of this same group were drinking an equivalent volume of fluid. Food and water were provided *at libitum* during the ten weeks of the experiment.

At the end of the supplementation period, mice in the postprandial state were anaesthetized with a 60 mg/kg intraperitoneal dose of sodium pentobarbital and euthanized by cervical dislocation. All experiments were conducted according to the Mexican Norm NOM-062-ZOO-1999 and approved by the Institutional Care and Use of Laboratory Animals Committee from Cinvestav-Mexico (CICUAL; protocol number 0236-17).

### 2.2. Cecal Microbiota Analysis

#### 2.2.1. Cecal Sample Collection and DNA Extraction

Cecal content was collected, snap frozen immediately in liquid nitrogen and stored at −70 °C until processing. DNA extraction was carried out using QIAamp PowerFecal DNA kit (Qiagen, Hilden, Germany), according to the instructions provided by the manufacturer. Samples were stored at −20 °C until analysis.

#### 2.2.2. Metagenomic Sequencing of Cecal Microbiota

Next generation sequencing library preparations and Illumina MiSeq sequencing were conducted at GENEWIZ, Inc. (South Plainfield, NJ, USA). All DNA samples were quantified using a Qubit 2.0 Fluorometer (Invitrogen, Grand Island, NY, USA). DNA (30–50 ng) was used to generate amplicons using a MetaVx^TM^ Library Preparation kit (GENEWIZ, Inc., South Plainfield, NJ, USA). The V3 and V4 hypervariable regions of prokaryotic 16S ribosomal DNA were selected for generating amplicons and subsequent taxonomy analyses. GENEWIZ designed a panel of proprietary primers targeting relatively conserved regions bordering the V3 and V4 hypervariable regions of bacteria 16S ribosomal DNA. The V3 and V4 regions were amplified using forward primers containing the sequence “CCTACGGRRBGCASCAGKVRVGAAT˝ and reverse primers containing the sequence “GGACTACNVGGGTWTCTAATCC˝. The first round PCR products were used as templates for second round amplicon enrichment by PCR. At the same time, indexed adapters were added to the ends of the 16S rDNA amplicons to generate indexed libraries ready for downstream NGS sequencing on an Illumina MiSeq.

DNA libraries were validated on an Agilent 2100 Bioanalyzer (Agilent Technologies, Palo Alto, CA, USA) and quantified with a Qubit 2.0 Fluorometer. DNA libraries were multiplexed and loaded on an Illumina MiSeq instrument according to manufacturer’s instructions (Illumina, San Diego, CA, USA). Sequencing was performed using a 2 × 300 paired-end (PE) configuration; image analysis and base calling were conducted with the MiSeq Control Software (MCS) embedded in the MiSeq instrument.

#### 2.2.3. Sequence Analysis

Sequences were processed and analyzed in the QIIME software package (Quantitative Insights Into Microbial Ecology, v1.9.1) [23]. The filtered sequence reads (Phred ≥ Q20) were employed to pick the operational taxonomic units (OTUs), with an open-reference OTU picking method based on 97% identity to entries in the Greengenes database (v13_8) by UCLUST [24] and PyNAST [25] alignment algorithms. Chimeric sequences were removed using ChimeraSlayer. We obtained a total of 2,379,361 million high-quality sequences, with an average of 158,624 ± 21,487 sequences per sample, from 15 cecal samples (*n* = 5/group). All communities were rarefied to 115,152 reads per sample to calculate bacterial diversity. Alpha diversity was calculated using Chao1 and Shannon indices. Chao1 index denotes the richness of the community (the estimated number of species/features per sample) and Shannon index denotes community diversity [26]. Alpha diversity between groups was compared using the Monte Carlo permutation procedure with 999 random permutations. Beta diversity was assessed by calculating weighted Unifrac distance matrices and then represented by two-dimensional principal coordinates analysis (PCoA) plot [27]. Results of the PCoA were then statistically tested by PERMANOVA with 999 permutations. In addition, Mahalanobis distance (Md) and Hotelling’s T^2^ test were applied to measure and confirm the differences between each pair of clusters [28].

Linear discriminant analysis effect size (LEfSe) was used to identify significant differences in relative abundance of bacterial taxonomy. The threshold for the logarithmic LDA score was 3.0 with the alpha value of 0.05 for the factorial Kruskal–Wallis test.

The raw sequences supporting the results in this article are available in the NCBI Sequence Read Archive repository under accession no. PRJNA605026.

### 2.3. Fecal Metabolites Extraction

Feces were collected from each mouse at week ten of the experiment, lyophilized, triturated, and homogenized to determine fecal metabolite profiles. Fecal metabolites extraction was carried out as previously reported [29]. Briefly, one-hundred mg of feces were extracted three times with chloroform/methanol (2:1), one mL each time. After that, the all extracts were combined and the solvent was evaporated. The residue was re-suspended in 1 mL of chloroform/methanol (2:1) and an aliquot of 50 µL was changed to a vial. The solvent in the aliquot was evaporated under nitrogen flux, then derivatized using N, O-Bis(trimethylsilyl)trifuoroacetamine with 1% trimethylchlorosilane (80 µL) and pyridine (20 µL) at 80 °C for 25 min. Once the system was at room temperature, isooctane was added to a final volume of 200 µL. Heptadecanoic acid was added to all vial as an internal standard at a final concentration of 3 mg/mL.

#### Gas Chromatograph-Mass Spectrometry Conditions and Fecal Metabolites Analysis

For GC/MS analysis one µL of the isoctane phase was injected in a pulsed-splitless mode. Injector temperature was set to 260 °C. A HP-5-MS capillary column (30 m × 25 µm × 0.25 µm) was used with helium as the carrier gas at constant flow rate of 1 mL/min. Oven program began at 40 °C (held 5 min), then increased at rate of 6 °C/min until 170 °C, then a second temperature ramp of 12 °C/min until 290 °C was applied. Transfer line temperature was set at 260 °C. Mass spectrometer operated at 70 eV of electron energy, quadrupole and ion-source temperatures were 150 and 230 °C, respectively. All data were obtained scanning from 40–550 *m*/*z*, while MassHunter Workstation software version B.0.0.6 (Agilent Technologies, Inc.) was used to collect all the data generated. Components mass spectra and retention times were obtained using the AMDIS (Automated Mass Spectral Deconvolution and Identification System) software.

Fecal metabolites analyses were implemented in the R 3.6.1 environment [30]. Raw data were normalized and transformed. A principal component analysis (PCA) was applied over the whole pre-processed dataset using the ade4 package [31]; differences between groups on PCA where confirmed by means of Mahalanobis distances (Md) for cluster separations and Hotelling’s T^2^ and F-test statistics for significances [28]. A hierarchical clustering analysis (HCA) was performed on the PCA patterns through FactoMineR packge [32]. Peaks with the lowest *p*-values on PCA and HCA were selected and annotated by comparing their respective extracted mass spectrum with the mass spectra of data of the NIST (National Institute of Standards and Technology, USA) library and software. A heatmap was created with all the relevant information obtained for the metabolites analysis.

### 2.4. Lipopolysaccharides, Hormones, and Cytokines Analysis

Upon mice sacrifice, portal blood was collected in tubes containing a dipeptidyl peptidase IV inhibitor (0.01 mL per mL of blood; Millipore, St. Louis, MO, USA) and centrifuged at 1600× *g* for 15 min at 4 °C. Serum was stored at −80 °C until analysis. LPS determinations were performed employing a mouse LPS kit (MyBioSource, San Diego, CA, USA) following the manufacturer’s instructions. Beside, GLP-1 (active), insulin, and leptin concentrations were quantified using a Mouse Diabetes Standard BioPlex Pro kit (Bio-Plex Pro Assay, Bio Rad, Hercules, CA, USA); while interleukins (IL-1α, IL-1β, IL-6, IL-10) and TNF-α concentrations were analyzed employing a Mouse Cytokine BioPlex Pro kit (Bio-Plex Pro Assay, Bio Rad, USA) and a BioPlex 200 instrument according to the manufacturer’s specification.

### 2.5. Morphological Analysis of Liver and Adipose Tissue

Liver and white adipose tissue (WAT) were fixed in 4% paraformaldehyde and processed for histology. The samples were sectioned in 10 µm slices in a rotatory retracting microtome (LKB, Sweden). The liver and WAT slices were stained with hematoxylin and eosin (H&E) solutions and permanently mounted on Entellan^®^ resin (Merck, Darmstadt, Germany). The sections were examined under a light microscope (BX50, Olympus, Tokyo, Japan). Besides, the WAT slices were cleaned and incubated in phosphate buffer during 15 min at room temperature; subsequently, they were stained with fresh solution (75% glycerol: 25% water; and 5% red Nile) of the stock solution of acetone-nile red [500 µg/mL] (Sigma-Adrich, St. Louis, MO, USA). The sections were observed in a fluorescence microscope at 530 nm of excitation wavelength and 550 nm of emission wavelength (BX50, Olympus, Japan).

#### Liver and Adipose Tissue Analysis by Fluorescence and Multiphoton Microscopy

Liver samples were fixed in a sucrose solution (300 mg/mL) dissolved in phosphate buffer and incubated during 3 h at 4 °C. Subsequently, the tissues were fixed in Leica tissue freezing medium (Leica Biosystems, United Kingdom) and cut in sections of 10 µm thickness. Sections were washed with phosphate buffer, stained with 10 µL of red Nile from a stock solution of acetone-Nile red [500 µg/mL], and 1 µL DAPI (Sigma-Adrich, St. Louis, MO, USA) dissolved in a fresh solution of glycerol at 75% (Sigma-Adrich, St. Louis, MO, USA) and incubated for 30 min. Finally, the sections were washed with distilled water and embedded in fluoroshield^TM^ mounting medium (Sigma-Adrich, St. Louis, MO, USA). Observations were performed using a multiphoton microscope system (LSM 880-NLO, Zeiss, Germany) coupled to an infrared laser Ti: Sapphire (Chameleon vision II, COHERENT, Scotland). The operating conditions in all experiments were Chameleon laser at 780 nm with intensity of 1.0% of power and open pinhole. Lipid droplets were observed with an immersion objective 40X//1.3, NA ∞−0.17, Zeiss Plan NEOFLUAR. Images were acquired by separation of the emission in two channels, DAPI (371–494 nm) for nucleus and red Nile (501–617 nm) for lipid droplets. All micrographs were captured in CZI format at 1131 × 1131 pixels and RGB format.

### 2.6. Body Weight Gain, Food Intake, Glucose, Triglyceride, and Cholesterol Determinations

Body weight and food intake were recorded every week throughout the whole experiment. Blood samples were taken in the postprandial state from the all mice tails in order to measure glucose, triglycerides, and cholesterol. Blood glucose concentrations were measured after taking the sample using a blood glucose meter (SD Check Gold, Mexico). Blood for triglyceride and cholesterol analysis was collected and centrifuged at 1600× *g* for 15 min at 4 °C. Serum was stored at −80 °C until analysis, which was carried out using kits coupling enzymatic reaction (BioVision, Milpitas, CA, USA).

### 2.7. Statistical Analysis

Results are given as mean ± SEM. Statistical significance was assessed employing a one-way ANOVA, followed by Tukey multiple comparison test. Pearson correlation was applied to assess the relationship between key bacterial genera with metabolites, inflammation, and systemic effects. Results were considered statistically significant at *p*-value < 0.05. When significant differences were found, they were indicated in figures by * *p* < 0.05, ** *p* < 0.01, and *** *p* < 0.001. Statistical analyses were performed using GraphPad Prism 8.0 (GraphPad Software, La Jolla, CA, USA).

## 3. Results

### 3.1. Agavins Supplementation Changed the Cecal Microbiota Composition and Fecal Metabolic High-Fat Diet-Fed Mice Profiling

In general, the mouse cecal microbiota of ST, HF, and HFA groups was considerably dominated by three phyla (Bacteroidetes, Firmicutes, and Proteobacteria) with six other minor phyla (Cyanobacteria, Actinobacteria, Deferribacteres, Tenericutes, Verrucomicrobia, and TM7; Figure 1a). Moreover, at phylum level, agavins intake led to substantial shifts on gut microbiota composition, showing similar relative abundance of dominant phyla than the ST group (Figure 1a). In comparison to HF group, HF+A increased approximately 33% the relative abundance of Bacteroidetes (S24_7 and Rikenellaceae families including *Bacteroides*, *Parabacteroides*, *Prevotella*, and *Odoribacter* genus) and decreased in about 33% the Firmicutes (Lachnospiraceae and Ruminococcaceae families including *AF12*, *Oscillospira*, *Allobaculum*, and *Ruminococcus* genus) and approximately 11% in the Proteobacteria phyla (Desulfovibrionaceae and Helicobacteraceae families; Figure 1b). In addition, the Firmicutes/Bacteroidetes ratio significantly decreased with prebiotic supplementation from 1.51 observed in HF group to 0.43 for HF+A group (*p* < 0.01); even reaching very close values to ST group (0.41; *p* > 0.05, Figure 1b). On the other hand, as expected, agavins consumption remarkably decreased bacterial richness and bacterial diversity in relation to ST and HF groups (*p* < 0.05; Appendix A). Since the intake of high-fat diets is associated with reduced bacterial diversity in the cecal microbiota [33]; in addition, prebiotics supplements usually promote the growth of specific bacterial taxa (conventional and new generation probiotics) [34,35]. Moreover, PCoA plot revealed that HF+A group had a distinct bacterial community structure, it is clustered separately from HF and ST groups, suggesting that prebiotic supplement affected gut microbial composition (Appendix A). Furthermore, HF+A group displayed a notable change on the cecal microbiota composition in relation to the HF, enriching the abundance of five desirable genera (*Bacteroides*, *Parabacteroides*, *Prevotella*, *Allobaculum*, and *Akkermansia*), while HF group increased eleven genera (*Oscillospira*, *Mucispirillum*, *Ruminococcus*, *Odoribacter*, *Flexispira*, *AF12*, *Bilophila*, *Desulfovibrio*, *Streptococcus*, *Roseburia*, and *Helicobacter*); LDA > 3.0; Figure 1d.

Agavins consumption not only induced alterations in the composition of the cecal microbiota, but also modified the fecal metabolites profiling (Figure 2). We detected 123 metabolites, of them, 34 showed the greatest differences between the treatments, these metabolites were selected for their lowest *p*-values in the multivariate analyzes. Among the 34 differentiated metabolites, most of them belong to the fatty acids family and their corresponding esters; sterols, carbohydrates, alcohols, amino acids, and some nucleobases were also observed. Nevertheless, only 24 compounds were annotated; however, mass spectra information allowed to propose the family to which four compounds belong to (Figure 2a).

High-fat diet consumption (alone or with the agavins supplement) increased the excretion of lanosterol and 1-O-octadcylglycerol as well as the acids: cis-13-eicosenoic, oleic, palmitelaidic, arachidic, and stearic; in relation to ST mice (Figure 2a). On the other hand, l-leucine, l-valine, uracil, thymine, octadecanoic acid, ethyl ester; n-pentadecanoic acid, and methyl-tetradecanoic acid were the metabolites depleted or notably decreased with HF diet, but recovered with agavins intake; therefore, some of those compounds could be used as potential biomarkers for agavins prebiotic consumption. Figure 2b presents a PCA showing a very clear and separated clusters between the ST mice group and animals fed with HF diets (alone or plus the agavins supplementation); besides, HF+A group is clearly separated into a distinct cluster from the HF group, confirming the differences observed in the fecal metabolomics profiling.

### 3.2. Agavins Intake Reduced LPS, Pro-Inflammatory Cytokines, Insulin, and Leptin Levels with Notable Increase of GLP-1 in Mice Fed High-Fat Diet

HF mice group exhibited the higher LPS concentration in relation to ST and HF+A mice groups (*p* < 0.001). Noticeably, HF+A group had a significant reduction of about 37% LPS levels compared to HF group (Figure 3a). On the other hand, we found that pro-inflammatory cytokines (IL-1α, IL-1β, IL-6, and TNF-α) evaluated in the present study showed a remarkably increment in HF in relation to ST and HF+A groups (Figure 3b–e). Interestingly, no significant differences on these pro-inflammatory cytokines levels were observed between ST and HF+A groups (*p* > 0.05). Noteworthy, agavins intake led obese mice to an important decrease of IL-1α (51%); IL-1β (42%); IL-6 (53%); and TNF-α (41%) concentrations when compared to HF mice (Figure 3b–e). Moreover, HF mice group exhibited the lowest concentration of the anti-inflammatory cytokine IL-10 compared to ST and HF+A groups; in contrast, HF+A mice showed a significant increment of this anti-inflammatory cytokine in relation to HF group (*p* < 0.05; Figure 3f).

On the other hand, HF mice were distinguished by exhibiting the lowest concentration of GLP-1 with respect to ST and HF+A; conversely, HF+A group showed a remarkable increment on GLP-1 levels in relation to ST and HF groups (*p* < 0.001; Figure 3g). Moreover, HF mice showed a significant increase of insulin and leptin levels compared to ST and HF+A (*p* < 0.01; Figure 3h–i). Noticeably, HF+A mice exhibited a remarkable reduction of about of 32% on insulin levels and 72% on leptin concentration with respect to HF group (Figure 3h–i).

### 3.3. Agavins Consumption Induced Morphological Changes on Cecum, Liver, and Adipose Tissue of Mice Fed High-Fat Diet

HF group showed a significant reduction of cecum full weight (0.21 ± 0.05 g) compared to ST and HF+A mice (0.55 ± 0.11 g and 0.44 ± 0.05 g, respectively; *p* < 0.001); in contrast, we did not observe significant differences in the cecum full weight between ST and HF+A groups (*p* > 0.05). On the other hand, the examination of HF liver evidenced a white dots pattern on the surface of the liver (Figure 4b, white arrow) and change of color due to accumulation of fat, which is a typical feature of fatty liver. Interestingly, HF+A group showed a recovery of the red color and a drastic decreased on the size of the fat dots (Figure 4c, white arrow). The hematoxylin and eosin (H&E) staining allowed us to compare the morphological changes in the hepatic tissue. The ST group displayed the typical morphology of a healthy hepatocyte, without lipid droplets or macrovesicular steatosis presence (Figure 4d). However, the HF liver tissue showed macrovesicular steatosis in hepatocytes located in the perisinusoidal zone (Figure 4e, black arrow). The hepatocytes with steatosis exhibited a large vesicle similar to lipid droplets, and the nucleus was displaced to the cell membrane, acquiring an elongated shape. Importantly, agavins consumption led the obese mice to a significantly reduction of the observed macrovesicular steatosis in the HF mice. Besides, we found small lipid droplets into the cytoplasm of some hepatocytes (Figure 4f, black arrow); moreover, the nucleus recovered its circular shape and its location within the cell (typically centrally located). Multiphoton microscopy observations allowed the visualization of accumulation of lipids droplets in the hepatocytes. Lipid droplets accumulation was not observed in the cytoplasm of the ST group (Figure 4g). However, in the HF group, a large number of dispersed small lipid droplets were observed in the cytoplasm (Figure 4h, white arrow). In contrast, HF+A group showed a drastic decrement on lipid droplets accumulations (Figure 4i).

Moreover, the amount of adipose tissue was significantly higher in HF mice when compared to ST (2.07 ± 0.30 g versus 0.37 ± 0.07 g; *p* < 0.001) and HF+A mice (2.07 ± 0.30 g versus 1.40 ± 0.45 g; *p* < 0.01). Agavins supplement led obese mice to a notable reduction on the amount of adipose tissue, of about 32%, with respect to the HF mice group (Figure 4j–l). Furthermore, since various studies have suggested a very close relationship between adipocyte hypertrophy, insulin resistance, and inflammation in obesity [36,37,38,39], we performed a histological test of the white adipose tissue (WAT). The histologic changes of WAT with most severe damage was observed in HF mice; the adipocytes showed a hypertrophy in comparison to ST group. The chronic inflammation of adipocyte cells produced a displacement of the nucleus to the periphery (Figure 4n, black arrow). Interestingly, the size of the adipocytes was lower in the HF+A group; this might be related to the reverted inflammation of the tissue when agavins were administrated; notably, the presence of macrophages infiltration (Figure 4o, black arrow) is an important source of inflammation in the WAT and vascular damage [40]. The affinity of red Nile staining for membrane lipids, allowed visualization of the thickness membrane of adipocytes of the HF group (Figure 4q, black arrow) respect to ST and HF+A mice. Moreover, it showed sinuosity and confirmed the differences on the sizes between treatments. Noticeably, HF+A group showed a tendency to decrease the adipocyte size and reduce membrane thickness in relation to HF group (Figure 4r).

### 3.4. Agavins Supplementation Notably Decreased Body Weight Gain, and Triglycerides but not Cholesterol Concentrations in Peripheral Blood of Mice Fed High-Fat Diet

As expected, mice that received the high-fat diets (HF and HF+A) increased significantly and progressively their body weight gain throughout the study in relation to control group (ST) mice (*p* < 0.001). However, mice that received the agavins supplement in their drinking water (HF+A) showed a significant reduction of 11% of body weight compared to HF group (*p* < 0.001; Figure 5a). Moreover, no significant differences were found in the water intake between HF and HF+A groups (4.6 ± 0.32 mL versus 4.9 ± 0.13 mL; *p* > 0.05), while the energy intake was notably higher in the two mice groups fed with high fat (HF and HF+A) compared to the ST group (*p* < 0.01). Nevertheless, a slight reduction in energy intake was observed in the mice group that received the agavins supplement compared to HF diet alone, but it was not significantly different (*p* > 0.05; Figure 5b).

Consumption of HF diet led mice to a high glucose increment (7.88 ± 0.23 mM), triglycerides (1.52 ± 0.22 mM), and cholesterol (4.45 ± 0.22 mM) concentrations in blood with respect to ST group (6.71 ± 0.12 mM), (0.94 ± 0.13 mM), and (2.66 ± 0.23 mM) respectively (Figure 5c–e). Noteworthy, mice receiving agavins in their drinking water (HF+A), did not showed significant differences neither in glucose nor triglycerides levels compared to ST group (*p* > 0.05). Besides, the HF+A group showed a decrement of 6% on glucose levels and a remarkable decreased 57% on triglycerides concentration in relation to HF group, whereas no significant differences were found on cholesterol levels between mice that received the HF diet with or without the prebiotic supplementation (*p* > 0.05; Figure 5c–e).

Figure 6 summarizes correlations between abundance of key bacterial genera with fecal metabolites, inflammatory biomarkers, hormones, and systemic effects. In general, several bacteria that were significantly enriched with HF diet positively correlated with pro-inflammatory cytokines, triglycerides, body weight, insulin, leptin, LPS, and glucose (Figure 6 and Appendix A). Remarkably, Desulfovibrionaceae family and *Helicobacter* and *Roseburia* genera (substantially enriched with the HF diet) showed a significant and positive correlation with all the variable mentioned before; while S24_7 family (abundant in the ST and HF+A groups) negatively correlated with all variables. In addition, the relationship between the increment of the anti-inflammatory cytokine IL-10 was positively correlated with family S24_7 and *Bacteroides* genus. Moreover, higher abundance of the genera *Bacteroides*, *Parabacteroides*, *Akkermansia* and *Allobaculum* were differentially enriched with the agavins supplement and positively correlated with GLP-1 levels. Whereas, l-leucine and l-valine had a positive correlation with the S24_7 family and *Prevotella* genus.

## 4. Discussion

Nowadays, obesity is a serious worldwide public health problem that in many occasions is accompanied by metabolic alterations which include hyperglycemia and insulin resistance, among others. Previously, it was demonstrated that highly branched neo-fructans (agavins) consumption could be a very good dietary strategy to prevent or mitigate some problems associated with the metabolic syndrome derived from high-fat diet consumption [15,16,17]. However, to our knowledge, there is not an integral study evidencing the effects of agavins on gut microbiota regulation and their metabolic products as well as their impact on metabolic endotoxemia and low-grade inflammation, playing both fundamental roles on the development of the known metabolic syndrome [2,3].

Interestingly, agavins supplementation induced substantial changes in the gut microbiota composition as well as in the microbial metabolite profiles, which strongly suggests a switch of microbial metabolism toward the utilization of indigestible carbohydrates; these new data could be the first mechanism approach explaining why agavins can counteract many of the detrimental effects of a diet with high fat content.

Furthermore, cecal microbiota analysis revealed that HF diet induced widespread changes at the phylum level, particularly an increase of F/B ratio, which was positively correlated with body weight (Appendix A); and also coinciding with previous studies that associated the F/B ratio as a marker of obesity [15,41]. In addition, the proportion of Proteobacteria (Desulfovibrionaceae, and Helicobacteraceae families) was also increased in the HF group. Proteobacteria phylum abundance positively correlated with LPS levels, which is in accordance with a previous research [42]. On the other hand, the HF group showed a significant increment mostly of Desulfovibrionaceae family (Figure 1d), and this event showed a positive association with obesity-related metabolic parameters; being consistent with early studies [43,44].

Moreover, HF diet consumption did enrich specific taxas such as *Bilophila*, *Desulfovibrio*, and *Helicobacter*; abundance of these genera has also been associated with metabolic disorders and inflammatory bowel disease [45,46]. Importantly, prebiotic supplementation reverted the negative changes originated by the HF diet intake, decreasing the F/B ratio and the proportion of Proteobacteria. Furthermore, agavins intake led to a notable enrichment of *Bacteroides*, *Parabacteroides*, *Prevotella*, *Allobaculum*, and *Akkermansia* genus (Figure 1). Interestingly, *Bacteroides* and *Prevotella* are involved in the degradation of plant polysaccharides [35,47,48]. Moreover, the abundance of *Prevotella* was negatively correlated with obesity-related metabolic parameters (Appendix A); which agrees with several previous works [35,47,48]. In addition, the abundance of the genera: *Bacteroides*, *Parabacteroides*, *Akkermansia*, and *Allobaculum* exhibited a significant and positive correlation with GLP-1 levels, while the *Bacteroides* genus also had a positive correlation with the anti-inflammatory cytokine IL-10. Previously, *Allobaculum* was associated with body weight loss and to an improvement of metabolic parameters [44,49]; whereas, the enrichment of *Akkermansia* showed a positive correlation with GLP-1 levels, lower LPS concentration in blood, reduction of inflammation markers, decrement of fat mass development, insulin resistance, and dyslipidaemia in obese mice [45,50,51]. Altogether, agavins supplement can regulate the gut microbiota toward a more healthy profile; moreover, all these results suggest a great beneficial contribution of each genus significantly enriched with the prebiotic supplement.

On the other hand, we observed an increment on the excretion of n-pentadecanoic acid, methyl-tetradecanoic acid, and octadecanoic acid, ethyl ester (metabolites identified as strong possible biomarkers for prebiotic administration) suggesting a decrement on the efficiency of intestinal absorption of these fatty acids when prebiotics are supplied; this event is proposed as a mechanism to enhance the lipid metabolism of host [43]. Additionally, the positive correlation of n-pentadecanoic acid with IL-10; as well as the negative association between the acids: n-pentadecanoic, methyl-tetradecanoic, and octadecanoic, ethyl ester with insulin, LPS, pro-inflammatory cytokines (IL1-α and IL1-β), leptin, and triglycerides levels, suggest that these fatty acids might be contributing to mitigate metabolic endotoxemia and low-grade inflammation in the HF+A group. However, our study was merely correlative, therefore these associations require further investigation. Moreover, the essential amino acids such l-leucine and l-valine have also been identified as possible biomarkers of prebiotic supplementation. Interestingly, l-leucine and l-valine were inversely correlated with the levels of LPS, leptin, insulin, pro-inflammatory cytokines, body weight, glucose, triglycerides, and cholesterol concentrations (Appendix A). A previous work showed that l-leucine and l-valine are involved in regulation of food intake through gut satiety hormones [52], while other investigations have evidenced that these essential amino acids are associated with body weight regulation, insulin, and glucose homeostasis [53,54]. In a very recent study, Wu and colleagues identified l-leucine, l-isoleucine and l-valine as biomarkers in plasma of pigs that received an inulin supplement [55]. These branched amino acids were lower after inulin intervention and positively associated to total cholesterol and glucose levels [55]; nonetheless, these results are disagreed with the present study, because we detected a high content of l-leucine and l-valine which inversely correlated with cholesterol and glucose levels. These discrepancies could be due to sample used to metabolites analysis. Wu and colleagues employed plasma and here were used feces. Hence, we do not know the levels of l-leucine and l-valine that reached the systemic circulation. On the other hand, pyrimidines like thymine and uracil, were metabolites also identified as biomarkers of agavins intake, since they have been reported as crucial compounds with key functions in cell physiology [56]. The abundance of thymine was positively associated with IL-10 levels, whereas thymine and uracil showed a negative correlation with body weight, leptin, IL1-α, TNF-α, and LPS concentrations (Appendix A). These results suggest that thymine and uracil might be contributing on the reduction of metabolic endotoxemia and low-grade inflammation observed in the HF+A group. However, further investigation is required to clarify the underlying mechanisms and the specific roles that potential new metabolites enriched after agavins prebiotic intake are playing on the health and wellness of host.

On the other hand, LPS leakage derived from the gut microbiota, into the portal blood is a well-established mechanism of metabolic endotoxemia that provokes liver inflammation [11]. Besides, the storage of lipids on hepatocytes requires cell compartmentalization; the lipid droplets on the liver is the physiological response to HF diet alteration in rodents [57]. Here, we showed that agavins consumption drastically reduced the lipid droplets number/abundance in the liver of mice, this result is concordance with a previous study [22]. The fact that agavins can reduce lipid droplets content in the liver might be due in part to a lower portal LPS concentration (Figure 3a), because LPS accelerates the progression of hepatic injuries [12], whereas decreasing LPS concentration could be a powerful strategy for the control of metabolic diseases [5].

Moreover, leptin is a hormone secreted in proportion to adipose size tissue [38,39]. In addition, IL-6 and TNF-α production by adipose tissue also increases with increasing cell size [37]. Interestingly, the notably reduction of leptin, insulin, IL-6, and TNF-α concentrations as well as the significant increment of IL-10 levels observed only in the HF+A mice group might be associated with the lower WAT and adipocyte size found in these animals compared to the HF group (Figure 4). Since leptin may induce insulin resistance [36,39], the significant decrement on leptin levels in HF+A mice may also contribute to amelioration of insulin resistance that we observed in these animals.

On the other hand, previous data suggest that a HF diet may deleteriously affect the secretion of gut hormones and normal post-prandial signaling, which could jolt on insulin secretion, satiety, and, finally, on body weight gain [58]. In the present work, we saw a reduction of GLP-1 levels in HF mice; nevertheless, this effect was reverted in the HF+A group. Increment of GLP-1 levels with the prebiotic consumption might be attributed to the increased number of L-cells (enteroendocrine cells producing GLP-1 in the jejunum and colon) as previously was reported for linear fructans [51] as well as a SCFA production derived from microbial fermentation of agavins [16,17]. In addition, levels of GLP-1 were positively associated with octadecanoic acid, ethyl ester and uracil concentrations (Appendix A). GLP-1 is an incretin hormone capable of promoting satiety [59]; then, the slight decrease on energy intake and the lower body weight gain observed in HF+A mice may be due in part to the notably increment of GLP-1 levels. Altogether, these results imply that agavins intake exhibits robust efficacy against metabolic syndrome in mice fed high-fat diet.

## 5. Conclusions

In the present study, we have shown that agavins supplementation mitigate metabolic endotoxemia and low-grade inflammation, in association with gut microbiota regulation and their metabolic products, inducing beneficial consequences on liver, adipose tissue, and metabolic disorders. Agavins intake reverted the microbiota changes caused by HF diet intake, reduced the Firmicutes/Bacteroidetes ratio and Proteobacteria phyla, which positively correlated with body weight and LPS levels; respectively. Remarkably, the genera *Bacteroides*, *Parabacteroides*, *Prevotella*, *Allobaculum*, and *Akkermansia*, enriched substantially with the agavins supplement, were associated with positive effects on host health.

On the other hand, l-leucine, l-valine, uracil, thymine, n-pentadecanoic acid, methyl-tetradecanoic acid, and octadecanoic acid, ethyl ester, metabolites identified as potential biomarkers for agavins prebiotic, showed an inverse correlation with LPS, pro-inflammatory cytokines, leptin, insulin, and triglycerides levels, suggesting that these metabolites might be helping to mitigate metabolic endotoxemia and low-grade inflammation; nonetheless, deeper investigations should be conducted to better understand the links we found between specific fecal metabolites and pathophysiological variables.

Moreover, the noticeably decrement of LPS concentration and pro-inflammatory cytokines, observed in mice that received the agavins supplement, could be contributing with the decrement of lipid droplets abundance in the liver, adipocyte size and leptin levels. Altogether, these changes induced an improve on glucose, triglycerides, and insulin levels as well as lower body weight gain of mice.

Finally, agavins can be a good dietary alternative or may be used on the development of novel therapies for the treatment of obesity and its associated comorbidities. However, clinical studies are needed to establish whether these agavins benefic health effects are also observed in humans.

## Figures and Tables

**Figure 1 foods-09-01792-f001:**
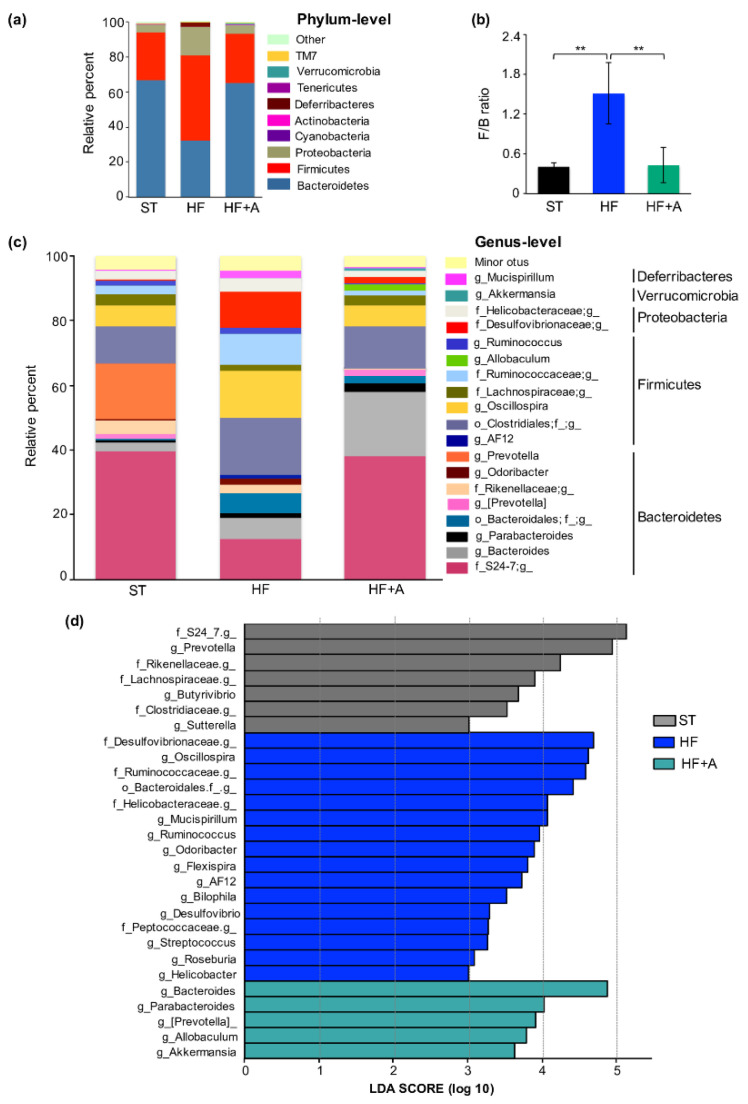
Differences in relative abundance of bacterial taxa in mice fed with high-fat diet plus agavins supplementation. (**a**) Bacterial taxa plot at the phylum level. (**b**) Firmicutes/Bacteroidetes ratio. Data are shown as average ± SD, and were analyzed using one-way ANOVA, followed by Tukey multiple comparison test. Significant difference is indicated by ** *p* < 0.01. (**c**) Bacterial taxa plot at the genus level. Each taxa > 1% of the average relative abundance in groups is indicated by a different color. Taxa are reported at the lowest identifiable level, indicated by the letter preceding the underscore: f, family; g, genus. (**d**) Histogram of biomarker bacteria in each group. Linear Discriminant Analysis (LDA) Effect Size (>3.0 fold) was used to determine statistically significant biomarkers. HF (mice fed with a high-fat diet); HF+A (mice fed with a high-fat diet plus agavins); ST (mice fed with a standard diet); for each experimental group (*n* = 5).

**Figure 2 foods-09-01792-f002:**
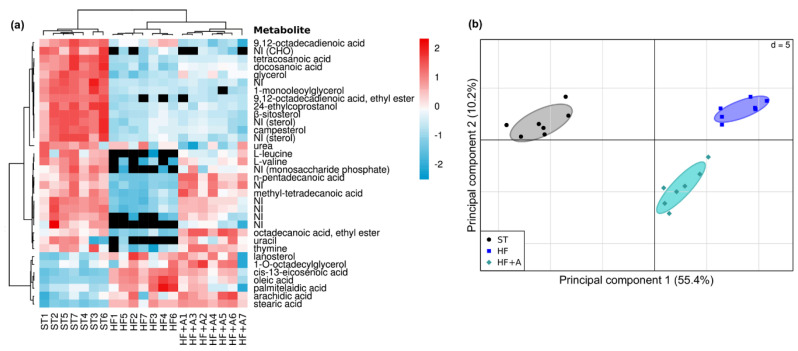
Fecal metabolites differences after agavins consumption. (**a**) Heat map of differential metabolites found in the feces of mice. NI (Not Identified); Colors (red, relative increase; blue, relative decrease; black, absence metabolite). (**b**) PCA plot showing separated clusters between the different treatments. Mahalanobis distance (Md) and Hotelling’s T^2^ test were calculated to measure and confirm the difference between every pair of clusters (ST vs. HF: Md = 372.8, *p* = 1.33 × 10^−9^; ST vs. HF+A: Md = 219.4, *p* = 9.89 × 10^−11^; HF vs. HF+A: Md = 91.3, *p* = 3.63 × 10^−6^). HF (mice fed with a high-fat diet); HF + A (mice fed with a high-fat diet plus agavins); ST (mice fed with a standard diet); for each experimental group (*n* = 7).

**Figure 3 foods-09-01792-f003:**
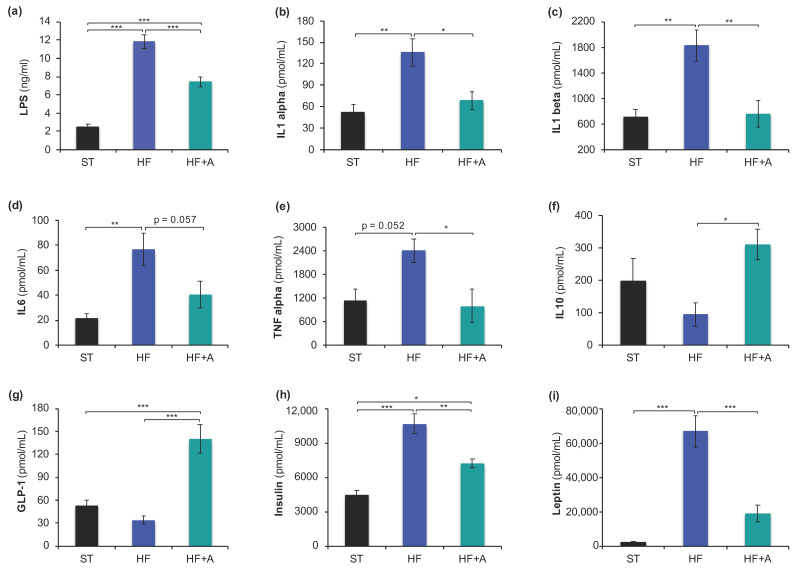
Agavins consumption decreased LPS and pro-inflammatory cytokine levels as well as insulin and leptin hormones in mice fed high-fat diet. (**a**) Concentration of LPS; (**b**) IL-1α; (**c**) IL-1β; (**d**) IL-6; (**e**) TNF-α; (**f**) IL-10; (**g**) GLP-1; (**h**) Insulin; and (**i**) Leptin. Data are shown as average ± SEM (*n* = 6/group). Data were analyzed using one-way ANOVA, followed by Tukey multiple comparison test. Significant difference is indicated by * *p* < 0.05, ** *p* < 0.01 and *** *p* < 0.001; near-significant differences are also reported. HF (mice fed with a high-fat diet); HF+A (mice fed with a high-fat diet plus agavins); ST (mice fed with a standard diet).

**Figure 4 foods-09-01792-f004:**
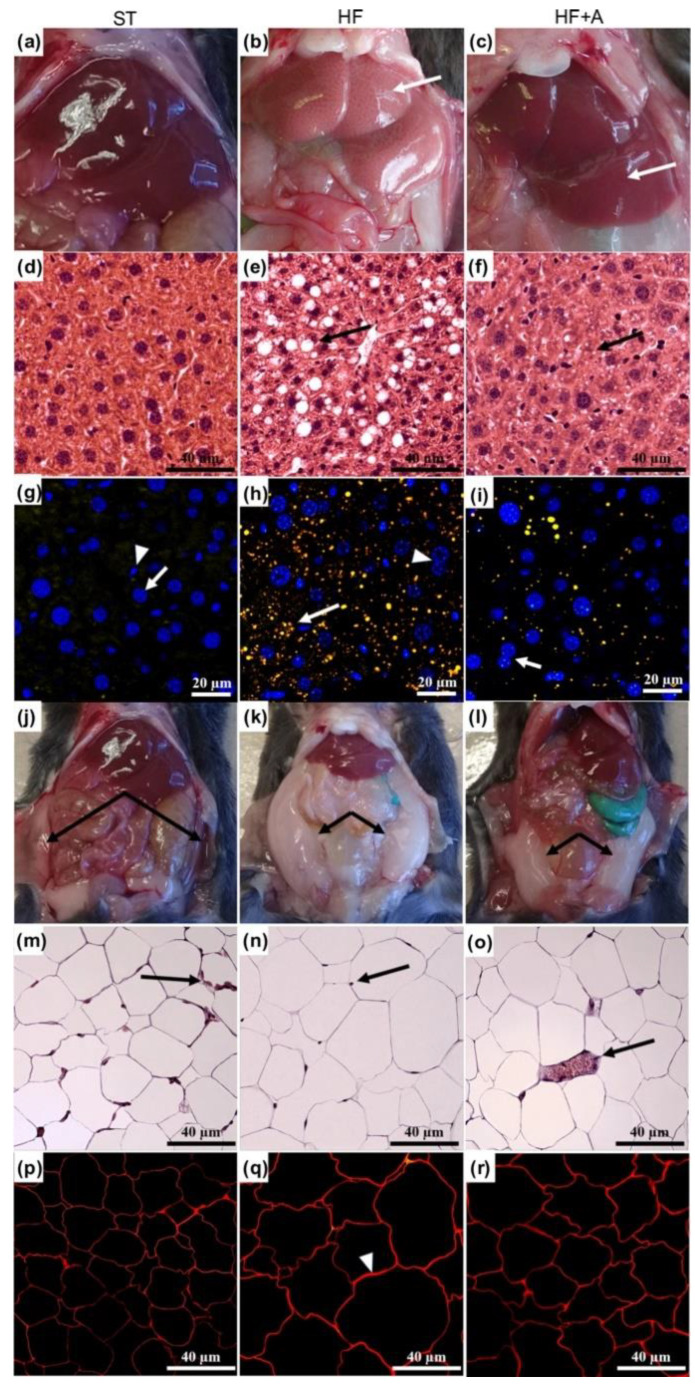
Agavins supplementation reduced hepatic steatosis and lipid droplets content as well as the amount of adipose tissue and adipocyte size in mice fed high-fat diet. (**a**–**c**) Representative images of liver after consumption of the different diets; (**d**–**f**) Histopathological examination of mouse livers by hematoxylin and eosin staining; (**g**–**i**) Accumulation of lipid droplets in the mouse livers. Lipid droplets were stained with red Nile (yellow) and nucleus with DAPI (blue); and analyzed using a multiphoton microscope system. (**j**–**l**) Representative images of adipose tissue present in the mice after the intake of the different diets; (**m**–**o**) Histopathological examination of adipocyte size using hematoxylin and eosin staining; (**p**–**r**) Adipocytes stained with red Nile. HF (mice fed with a high-fat diet); HF+A (mice fed with a high-fat diet plus agavins); ST (mice fed with a standard diet).

**Figure 5 foods-09-01792-f005:**
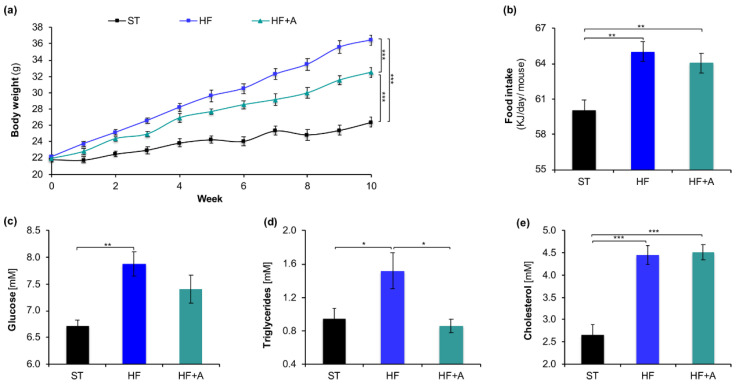
Agavins supplementation reduced body weight gain, food intake, glucose and cholesterol levels in mice fed high-fat diet. (**a**) Body weight evolution; (**b**) Food intake; (**c**) Glucose; (**d**) Triglycerides; (**e**) Cholesterol levels in blood. Data are shown as average ± SEM (*n* = 7/group). Data were analyzed using one-way ANOVA, followed by Tukey multiple comparison test. Significant difference is indicated by * *p* < 0.05, ** *p* < 0.01, *** *p* < 0.001. HF (mice fed with a high-fat diet); HF+A (mice fed with a high-fat diet plus agavins); ST (mice fed with a standard diet).

**Figure 6 foods-09-01792-f006:**
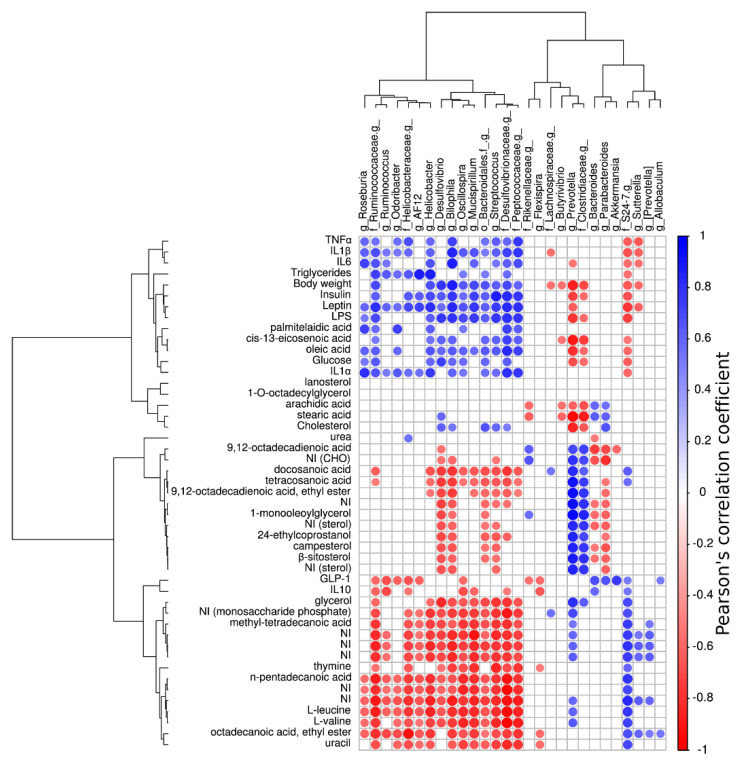
Correlogram showing only significant Pearson’s correlations between key bacterial genera and metabolites, inflammatory biomarkers, hormones, and systemic effects. Blue circles denote a positive correlation while red ones denote a negative correlation.

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
