# Peer review of "Highly Branched Neo-Fructans (Agavins) Attenuate Metabolic Endotoxemia and Low-Grade Inflammation in Association with Gut Microbiota Modulation on High-Fat Diet-Fed Mice"

_foods, 2020, doi:10.3390/foods9121792_

Round 1

Reviewer 1 Report

This study describes the effects of Agavins on metabolic endotoxemia and low-grade inflammation through gut microbiota modulation and metabolite changes in high-fat-diet fed mice. The experiments were well organized, and the manuscript was well written as well. I believe this work is good enough. I only have some minor questions.

  1. Line 194-202: It seems metabolites were identified without authentic standards. I have some concern about the reliability of identified compounds.
  2. Figure 1(d): Please add information on which color denotes which group.
  3. Line 294-295: Like mentioned above, how were metabolites "completely identified" without using authentic standards and/or NMR? Please check references widely accepted for metabolite identification with different levels (level I - level IV; Bino et al., 2004, Sumner et al., 2007, etc.).
  4. Figure 2(a) What does NI mean? Not identified?
  5. Figure 6: I am wondering why the relationships between metabolites and other parameters (e.g. inflammatory biomarkers) were not investigated. Metabolites might also be significantly correlated with these parameters like gut microbiota.
  6. Minor spelling errors (e.g. Line 176: tree times).

Author Response

This study describes the effects of Agavins on metabolic endotoxemia and low-grade inflammation through gut microbiota modulation and metabolite changes in high-fat-diet fed mice. The experiments were well organized, and the manuscript was well written as well. I believe this work is good enough. I only have some minor questions.

  • Line 194-202: It seems metabolites were identified without authentic standards. I have some concern about the reliability of identified compounds.

R: In a significant proportion of untargeted metabolomics research studies, the peaks are detected and searched against databases; rarely do investigators undertake the challenging and time consuming step of confirming the putative detected peaks using authentic standards due to the vast structural heterogeneity of metabolites as well as their large number (Zeki et al. 2020; Viant et al. 2017). Therefore, we can compare similarity of fragmentation patterns of “putatively annotated compounds” with public or commercial libraries with a Level 2 of confidence, according to Chemical Analysis Working Group of the Metabolomics Standards Initiative (MSI). However, at level 2, only annotations are reached (Sumner et al. 2007). So, to be more precise, in the line 200 we change the word "identified" by "annotated".

Line 200: “and annotated by comparing their respective extracted mass spectrum…”

  • Figure 1(d): Please add information on which color denotes which group.

R: Thank you for pointing this out. We have added this information to Figure 1(d).

  • Line 294-295: Like mentioned above, how were metabolites "completely identified" without using authentic standards and/or NMR? Please check references widely accepted for metabolite identification with different levels (level I - level IV; Bino et al., 2004, Sumner et al., 2007, etc.).

R: Since, we carried out the comparison of metabolite mass spectrum with the mass spectra of data of the NIST library and software, only annotations were achieved. Therefore, to be more precise we have changed the sentence “completely identified” by “annotated”.

Line 297: “Nevertheless, only 24 compounds were annotated;...”

  • Figure 2(a) What does NI mean? Not identified?

R: Yes, NI means Not Identified. We have added this information to Figure 2(a).

  • Figure 6: I am wondering why the relationships between metabolites and other parameters (e.g. inflammatory biomarkers) were not investigated. Metabolites might also be significantly correlated with these parameters like gut microbiota.

R: Figure 6 is a summary of correlations between key bacterial genera with fecal metabolites, inflammatory biomarkers, hormones, and systemic effects. The relationships between metabolites and all metabolic parameters evaluated in the present study are shown in the Supplementary Figure 2, and discussed in the lines: 491-493, 498-503 and 512-516.

  • Minor spelling errors (e.g. Line 176: tree times).

R: Thank you for pointing this out. This is now corrected.

Reviewer 2 Report

The paper by Huazano-Garcia et al is focused on the examination of the influence on gut microbiota composition of rodents after the supplementation with agavins. The paper is well written and need to address one major comment given below.

1) the authors should discuss in the metaled manner the chemical composition of agavins given to animals - did the authors characterized the composition of agavins included in the diet? it is possible? In my opinion this info is crucial for the repeatability of the study by other groups

Author Response

The paper by Huazano-Garcia et al is focused on the examination of the influence on gut microbiota composition of rodents after the supplementation with agavins. The paper is well written and need to address one major comment given below.

the authors should discuss in the metaled manner the chemical composition of agavins given to animals - did the authors characterized the composition of agavins included in the diet? it is possible? In my opinion this info is crucial for the repeatability of the study by other groups

The chemical composition of agavins is made principally of fructose molecules linked by common β(2-1) like inulins plus β(2-6) linkages as well as the presence of an internal glucose unit. Agavins coming from Agave tequilana (BIOAGAVETM) used in the present study were purchased to Ingredion-Mexico. Ingredion is a global company with presence in more than 120 countries. Ingredion is characterized by their product and according to information provided by the supplier, each 100g of BIOAGAVETM contains 91.6 g of agavins (soluble fiber) and 2.8 g of sugars as well as 5.6% of moisture. We characterized the BIOAGAVETM product through HPAEC-PAD chromatography. The chromatographic profile obtained coincided with the well-established chromatographic patterns and characteristics for this type of branched fructans (data not shown).

Reviewer 3 Report

In my opinion, this work was very well planned and described, and the research was carried out properly. The publication was properly formatted.

Author Response

In my opinion, this work was very well planned and described, and the research was carried out properly. The publication was properly formatted.

R: Thank you for this comment.

Round 2

Reviewer 2 Report

The authors should include chromatogram of BIOAGAVE as supplementary materials.

Author Response

The authors should include chromatogram of BIOAGAVE as supplementary materials.

R: We have added this information as supplementary material. See Figure 1S.